# Association between the Expression of Sexual Orientation and/or Gender Identity and Mental Health Perceptions in the Peruvian LGBTI Population

**DOI:** 10.3390/ijerph20095655

**Published:** 2023-04-27

**Authors:** Jane Castaneda, Nicanor Poma, Benoit Mougenot, Percy Herrera-Añazco

**Affiliations:** 1Medical School, Universidad Peruana de Ciencias Aplicadas, Lima 15023, Peru; 2Centro de Excelencia en Investigaciones Económicas y Sociales en Salud, Universidad San Ignacio de Loyola, Lima 15024, Peru; 3Facultad de Ciencias Empresariales, Universidad San Ignacio de Loyola, Lima 15024, Peru; 4Red Internacional en Salud Colectiva y Salud Intercultural, Mexico City 56900, Mexico; 5Instituto de Evaluación de Tecnologías en Salud e Investigación, Seguro Social de Salud, Lima 14072, Peru

**Keywords:** depression, anxiety, sexual orientation, gender identity, LGBTI people

## Abstract

Introduction: The non-expression of sexual orientation and gender identity can affect mental health in the lesbian, gay, bisexual, transgender, and intersex population in Peru. Method: Secondary, observational, analytical, and cross-sectional analyses of data from the “First Virtual Survey on the LGBTI population” were conducted with a population (*n* = 11,345) of LGBTI adults aged 18 years old or more. The variables of mental health and expression of sexual orientation and/or gender identity were measured using a self-reported questionnaire that did not include a validated scale; questions with multiple alternatives that included “yes” and “no” options were used. Prevalence ratios (PR) and 95% confidence intervals (95% CI) were obtained by glm log Poisson regression models. Results: The median age of the participants was 25 years (IQR: 21–30), and the majority of the population identified as gay, followed by lesbian and bisexual. Individuals who expressed their sexual orientation and/or gender identity were 17% less likely to have had perceived mental health problems in the last 12 months (PR: 0.83, 95% CI: 0.76–0.90, *p* < 0.001). Conclusions: The non-expression of sexual orientation and/or gender identity has a significant negative effect on the mental health problems of the LGBTI population. These results highlight the importance of promoting the expression of sexual orientation and gender identity in our community.

## 1. Introduction

While the demographics of the lesbian, gay, bisexual, transgender, transsexual, and intersex (LGBTI) community are incomplete [1], approximately 0.5% of men and 1% of women identify themselves as bisexual and 2% of men and 0.5% of women identify as fully homosexual in western countries [2]. In 2017, a study on the LGBTI community in Peru found that 35.2% stated that they were gay, 27.4% bisexual, and 21.4% lesbian. Likewise, 83.8% identified themselves as non-trans, 7.5% as non-binary gender people, 3% as trans males and 2% as trans females [3]. By 2020, 8% of Peruvians over 18 years of age considered themselves to be non-heterosexual [4].

Sex assignment at birth is based on sex chromosomes, genital anatomy, and hormone levels [5]. Gender, on the other hand, is determined by the behavioral characteristics or psychological aspects associated with sex that may vary with culture and over time [6]. Gender identity refers to the way one feels or thinks about one’s gender [7], which may or may not correspond to sex, and is independent of sexual orientation [8]. Sexual orientation refers to emotional, physical, romantic, sexual, and spiritual attraction, desire, or affection for another individual [8]. The disclosure of sexual orientation is the process of communicating and expressing same-sex attraction, commonly known as “coming out” [9]. Gender expression refers to the outward signs or ways in which a person expresses their gender [5]; that is, how one manifests feeling female, male, neither, or both, through behavior [6].

The LGBTI community tends to deal with discrimination due to stigmatization, criminalization, and to restrictions on the expression of their sexual and gender diversity [10,11]. The conception that non-normative expressions or identities are dangerous or that they pose a threat to moral and public order, as well as discrimination and intolerance, perpetuate violence against this community [12,13]. Therefore, some keep a part of their private lives secret and repress their sexual orientation or gender identity [14,15,16,17], which represents a significant source of stress for this community [18]. According to Meyer’s minority stress theory, chronic exposure to these stressors may be responsible for an increased risk of mental health problems in this community [19].

Mental health, according to the World Health Organization, is defined as a state of mental well-being that allows people to face stressful moments, fully develop their abilities, learn and work appropriately, and contribute to their community [20]. It is also considered a basic human right and is essential for personal, community and socio-economic development [20], and should not be considered solely as the absence of a mental illness or condition [21]. Although there are no standardized measures to measure good mental health, different domains have been proposed that can define it, among which we can find knowledge about mental health, attitudes towards mental health problems, self-perception and values, cognitive abilities, academic and work performance, emotions, behavior, individual coping strategies, social skills, family relationships, physical health, sexual health, meaning and quality of life [21]. Mental health problems can be mental disorders, psychosocial disabilities, and other effects associated with great distress, functional disability, or risk of self-injurious behavior [22]. Mental disorders are characterized by clinically significant cognitive impairment, and impaired emotional regulation or behavior in a person. These disorders are diverse, and among the most common are depression and anxiety [22]. Depression is characterized by a general feeling of sadness, anhedonia, apathy, worthlessness, and hopelessness [23]. Anxiety disorders are characterized by excessive fear and worry, and other associated behavioral disturbances [22].

Even though information on the mental health of the lesbian, gay and bisexual (LGB) population is inconclusive [24]. Studies in England, the United States and New Zealand show a higher prevalence of mental health problems in LGBTI people than in heterosexuals, such as depression, anxiety, suicidal thoughts and attempts, obsessive compulsive disorder, etc. [19,24,25,26,27]. For example, it was shown that the prevalence of depression and anxiety was 1.5 times higher in LGB people than in heterosexuals [24]. Similarly, in Peru, some research also suggests that there are mental health problems in this population. It has been observed that more than half of the LGBTI community presented mental health problems such as anxiety and depression [3]. A study found that non-binary gender was associated with more mental health problems compared to the non-trans population, and that being bisexual was associated with more mental health problems than gay people [28]; the factors associated with these problems were being female, not having a partner, and not having a job. Other research found that anxiety, low self-esteem, and depression were the most frequent mental health problems, followed by bipolar disorder, post-traumatic stress disorder, borderline personality disorder, and substance abuse [29]. Additionally, an investigation reported that the prevalence of anxiety and/or depression in the Peruvian LGBTI population was 23.8% [30].

The expression of sexual identity presents mixed effects on mental health [31], as it contributes to reducing the stress of hiding one’s sexual orientation but exposes the person to prejudice and discrimination [32]. Therefore, the results of the studies with LGBTI persons are discrepant: some researchers have indicated that those who express their sexual orientation present with fewer mental health problems [9,33,34], whereas other researchers have found the opposite [35,36,37]. These differences can be explained by the different ways in which the expression of sexual orientation and mental health is studied. However, there are sociodemographic and psychological factors [36,38] that prevent their comparison to contexts, as observed in Peru, where discrimination can reach 75% of the lesbian, gay, bisexual, trans or non-binary gender community [29] or 63% of the LGBTI population [3]. Therefore, the present study aimed to determine the presence of an association between the expression of sexual orientation and/or gender identity and mental health problems in the LGBTI population in Peru. In addition, it aimed to describe the sociodemographic characteristics of the target population and determine the prevalence of self-reported mental health problems and expression of sexual orientation and gender identity. Finally, improvement proposals were presented for future national surveys based on the LGBTI population to improve the representativeness of the sample and the reliability of the data collected.

## 2. Materials and Methods

### 2.1. Study Design

This observational, analytical and cross-sectional study used the database of the “First Virtual Survey on the LGBTI population”, conducted by the National Institute of Statistics and Informatics (INEI) between May and August 2017. Available online: http://iinei.inei.gob.pe/microdatos/Consulta_por_Encuesta.asp (accessed on 1 December 2022) as the secondary source. The survey contained 71 questions divided into six areas: sociodemographic characteristics (education, health, identity, body and sexuality, family environment, disability, employment, ethnicity); discrimination and violence; knowledge of LGBTI people; citizen participation; perception of LGBTI status; and housing and household data. As there was no previous information on LGBTI population size, non-probabilistic convenience sampling was used to recruit participants. Likewise, for the development of the survey, the LGBTI population was contacted through the main LGBTI organizations in Peru. The survey was available in Spanish and virtually through the INEI website [3].

### 2.2. Population, Sample, and Sampling

The present study included people aged 18 years or older who considered themselves LGBTI or who, without identifying themselves with these categories, did not ascribe to the binary definitions of masculine or feminine within the Peruvian territory. The proportion of participants who did not respond to the outcome variable or who self-identified as heterosexual was 5.7%. The final database of this research is BD_FinalLGBTI. Available online: https://osf.io/x7c64/?view_only=8e59cc5692884c1dbe17d3a6a3df045f (accessed on 1 December 2022).

### 2.3. Variables

The independent variable was the expression of sexual identity and/or gender identity, which was determined by question: “Do you express your sexual orientation and/or gender identity without fear?”. This variable was titled “Expression of sexual orientation and/or gender identity” and was assigned two categories: “Yes” and “No.”

The dependent variable was mental health, which was determined by the question: “In the last 12 months, have you faced health problems such as depression or anxiety? “. This variable was titled “Mental health problems” and was assigned two categories: “Yes” and “No.”

The following variables were additionally included: age, place of birth, place of origin, education, employment, ethnicity, disability, insurance, sex, sexual orientation, gender identity, family respect, acceptance and integration, relationship status, and discrimination and/or violence. The variable age was determined in years. The variable place of birth was determined by place of birth and was categorized into two groups: “Lima” and “Others”. The variable place of origin was determined by the geographical location of their home and was categorized into two groups: “Lima” and “Others”. The education variable was determined using the question: What is the last level of studies reached? and it was categorized into six groups: “No educational level”, “Initial”, “Primary”, “Secondary”, “Superior”, and “Postgraduate”. The employment variable was determined using the question: “Last week, did you work at least one hour for any payment in money or kind?”, and two categories were assigned: “Yes” and “No”. The ethnicity variable was determined using the question: “By your customs and your ancestors, do you feel or consider yourself?”, and it was categorized into three groups: “white”, “Mixed race” and “Ethnic minorities”. The disability variable was determined using the question: “Do you have any permanent disability or difficulty that prevents you from carrying out your daily activities normally, like other people?”, and two categories were assigned: “Yes” and “No”. The insurance variable was determined by the question: “Are you affiliated with some type of health insurance?” and two categories were assigned: “Yes” and “No”.

The sex variable was determined by the question: “Which sex were you registered with at birth?”, and two categories were assigned: “Female” and “Male”. The sexual orientation variable was determined by the statement: “According to your sexual orientation, do you currently consider yourself to be: “, and the variable was assigned six categories: “Gay”, “Lesbian”, “Bisexual”, “Pansexual”, “Asexual”, and “Other”. Those participants who reported being heterosexual were eliminated. The gender identity variable was determined by the statement: “According to your gender identity, do you consider yourself to be: (Select only one option) “, and was assigned six categories: “Trans female”, “Trans male”, “Non-binary gender person”, “Non-trans” and “Other.” The variable respect acceptance of and integration in the family was determined by question: “Upon learning about your sexual orientation/gender identity, did your family members respect, accept, and integrate you?”, and two categories were assigned: “Yes” and “No.” The variable relationship status was determined by the statement: “Currently in relation to your life as a couple, are you” and four categories were assigned: “Without a partner”, “With a partner” and “With more than one partner”. Finally, the variable discrimination was determined by question: “Have you ever suffered discrimination and/or violence?”, and two categories were assigned: “Yes” and “No.”

### 2.4. Statistical Analysis

For the analysis, the survey database was downloaded in SPSS format from the INEI web platform and analyzed using the STATA 17.0 ^®^ program (StataCorp LP, College Station, TX, USA). The categorical variables were described using absolute frequencies and percentages with their respective 95% confidence intervals (CI). The Shapiro–Wilk test was used to evaluate the normality of the age variable, and the Levene test was used to evaluate its homogeneity. Given that the age variable did not have a normal distribution, it was described using median and interquartile range (IQR) and the relationship of the age variable in the group that presented mental health problems and in the group that did not present mental health problems was evaluated using the Mann–Whitney U test. Bivariate analysis of the population characteristics according to the presence or absence of mental health problems was performed using the Chi-square test.

To determine the association between the expression of sexual orientation and/or gender identity and mental health, generalized linear models of the Poisson family were constructed with robust variance and log link, and prevalence ratios (PR) with their respective 95% confidence intervals (CI) were calculated. A crude Poisson regression model and one adjusted model were constructed. Statistical criteria were used to determine the variables that were included in the adjusted model. Variance inflation factors (VIF) were determined to evaluate the collinearity between variables in the adjusted models. A VIF of 10 was considered as the cut-off point [39].

## 3. Results

### 3.1. Population Characteristics

The population consisted of 12,026 individuals, of which 11,345 were included (Figure 1). The median age of the participants was 25 years (IQR: 21–30). Most of the participants were from the Lima region (69.4%), had a university education level (77.17%), and reported no disability (96.90%) (Table 1).

Of the study population, 53.11% were assigned male sex at birth. Regarding sexual orientation, 43.20% indicated that they were gay, 22.45% were lesbian, 26.03% were bisexual, 5.26% were pansexual, 0.68% were asexual and 2.37% identified themselves in another category. Regarding gender identity, 88.36% identified themselves as non-trans, 7.32% as non-binary gender, 2.06% as trans female, 2.12% as trans male, and 0.13% assigned themselves to another category. Regarding the expression of sexual orientation and/or gender identity, 58.59% did not express their sexual orientation and/or gender identity without fear, and 23.46% reported having presented any mental health problem in the last 12 months (Table 2).

### 3.2. Bivariate Analysis According to the Presence or Not of Mental Health Problems

The median age of individuals reporting no mental health problems was 25, and the median age of people reporting mental health problems was 23 Mann–Whitney U test *p* < 0.001. People who did not express their sexual orientation and/or gender identity presented a higher prevalence of mental health problems than those who did (*p* < 0.001). Likewise, those who did not report respect, acceptance, and family integration had a higher prevalence of mental health problems than those who reported them (*p* < 0.001). It was found that those who were discriminated against had a higher prevalence of mental health problems than those who had not (*p* < 0.001). Similarly, those assigned female sex at birth presented a higher prevalence of mental health problems than those assigned male sex at birth (*p* < 0.001) (Table 2).

### 3.3. Association between the Expression of Sexual Orientation and/or Gender Identity and Mental Health

In the crude model, a statistically significant association was found between the expression of sexual identity and/or gender identity and perceived mental health problems. Individuals that expressed their sexual orientation and/or gender identity without fear were 24% less likely to present perceived mental health problems during the last 12 months (PR: 0.76, 95% CI: 0.71–0.82, *p* < 0.001). In the first adjusted model, the initial association was maintained and was statistically significant; in this case, those who expressed their sexual orientation and/or gender identity without fear were 21% less likely to present perceived mental health problems during the last 12 months (PR: 0.79, 95% CI: 0.74–0.85, *p* < 0.001). Likewise, in the second adjusted model, the association was maintained and was also statistically significant, which revealed that those who expressed their sexual orientation and/or gender identity without fear were 17% less likely to present perceived mental health problems during the last 12 months (PR: 0.83, 95% CI: 0.76–0.90, *p* < 0.001). Similarly, a significant association was found between the variables age, employment, disability, insurance, sexual orientation, respect, acceptance and family integration and discrimination, and mental health problems. In a further analysis, no collinearity was found between the data (Table 3).

## 4. Discussion

Our findings revealed that LGBTI individuals who expressed their sexual orientation and/or gender identity without fear were less likely to report having experienced perceived mental health problems.

These results are similar to those found in other studies. A Californian study on lesbian, gay, and bisexual (LGB) youths found that all participants who expressed their sexual identity reported lower rates of depression, anxiety, and suicidal ideation [9]. In Buffalo, New York, another study, which also involved LGB youths, found that those who revealed their sexual identity presented a lower incidence of depression [40]. In Massachusetts, the non-disclosure of sexual orientation was found to be associated with more than 15 days of depression in the past month in lesbian and bisexual women but not in gay or bisexual men [34]. Research on lesbian women has shown that those who had higher levels of expression of their sexual orientation had less anxiety and depression [41]. Another study involving lesbian women found that the greater the expression of sexual orientation, the lower the anxiety levels [42]. Similarly, another group of researchers, using lesbian, bisexual and queer women as participants, found that greater expression of sexual orientation was associated with greater social support which, in turn, was independently associated with better mental health [43]. Finally, in Canada, LGB people who disclosed their sexual orientation had fewer symptoms of anxiety and depression [44].

However, other studies have found contradictory results. A study conducted in Boston found that gay people that expressed their sexual orientation, including anger, presented more depressed symptoms than those who expressed it and were integrated into the heterosexual community [45]. However, no association was found between any degree of non-expression and mental health problems [45]. On the other hand, in bisexual men in New York, it was found that greater disclosure of their sexual orientation was not related to depressive symptoms, anxiousness, or fewer positive emotions and expressions [33]. In research based on a California LGB population, it was observed that male-assigned individuals who did not express their sexual orientation were less likely to be depressed than those who expressed it [36]. Nonetheless, female-assigned individuals who expressed their sexual orientation were less likely to report major depressive disorder than those who did not express their sexual orientation [36]. One research from the United States on transgender individuals found that gender identity disclosure was not directly related to depression [46]. In the Netherlands, it was found that in LGB individuals, the degree of expression of sexual orientation was not significantly related to mental health [47]. Finally, a German study on gay men found that disclosure of sexual orientation had no relevant effect on mental health [48].

These results differed from ours in several aspects. In some cases, different questionnaires were used to evaluate the expression of sexual orientation [33,36,45,47,48] or the expression of gender identity [46]. Likewise, in some studies only the population assigned with the male sex at birth was included [33,45,48], only transgender [46], or an older population [46]. Similarly, unlike other studies, this study did not divide the population according to sex [36,47]. Another important aspect that differentiates this study from previous ones is the way in which the exposure variable was defined. This was understood in most studies as the disclosure of sexual orientation [36,43,45,47,48] and in others as the disclosure of gender identity [46]. In contrast, in our research, we jointly asked about the “fearless expression of sexual orientation and/or gender identity”. In addition, it is not certain that the “expression without fear” as asked in the survey, only fits the terms of disclosure.

The LGBTI community is exposed to multiple stressors, which makes them vulnerable to mental health problems [45] and, as we have shown, different studies have found contradictory associations between non-expression of sexual identity and mental health [36]. Nonetheless, our study, which is based on the subjective understanding of the free expression of sexual orientation and/or gender identity of members of the Peruvian LGBTI community, has revealed that the fear to freely express one’s sexual orientation and/or gender identity may lead individuals to deprive themselves of positive mental health outcomes.

This may be because members of the LGBTI community who are afraid of being stigmatized or violated for expressing their sexual orientation and/or gender identity [45] suffer from stress and anxiety by being aware of their behavior and the perception that their environment has of them [18]. This fear of freely expressing their sexual orientation and/or gender identity has its origin in the high level of discrimination perceived by the LGTB community [14]. In Peru, it is reported that discrimination affects more than 60% of this community [3,29,49], unlike other countries, where it can affect 33% [50]. In this sense, a Peruvian study found that LGBTI participants who were discriminated against were 10% more likely to have mental health problems [51].

When the LGBTI community internalizes negative attitudes, beliefs and stereotypes associated with non-normative gender identities and expressions (internalized heterosexism), a source of chronic stress is generated in association with negative mental health outcomes related to their self-acceptance [19,46,52]. On the other hand, those who freely express their sexual orientation or gender identity could present a decrease in stress and an improvement in their self-esteem [42,46]. In this sense, the disclosure of a stigmatized identity, such as the non-heteronormative one, can provide access to coping resources and improve internalized heterosexism [46]. Another aspect that could indirectly explain our results is that the study population was, for the most part, recruited thanks to LGBTI organizations. This could mean that these people have a support group, and this positively influences their mental health by reducing the potential risk factors to which they are exposed when expressing their sexual orientation or gender identity [9].

### 4.1. Implications for Public Health and Clinical Practice

In Peru, 61.9% of the LGBTI community perceived prejudice when going to the health center and being cared for by mental health personnel [29]. This means that 59.4% of this community does not want to consult a psychologist or psychiatrist due to bad experiences with health personnel [29]. This creates a barrier between the community of lesbian, gay, trans, bisexual, queer, and other sexual identities (LGBTQ +) and the health system, which prevents the addressing of mental health problems of this community [29]. Globally, mental health disorders represent the second leading cause of disability-adjusted life years and years of life lived with disabilities, accounting for between 12% and 31% of the global burden of disease, respectively [53].

In this regard, public policies could be implemented in favor of the right to health in this community that consider the importance of the expression of sexual orientation and gender identity [11]. In addition, the creation of support groups for the LGBTI community should be promoted, and their participation should be encouraged as they can help reduce the effects of stress on these minorities through the adoption of collective identity, disclosure of sexual orientation to others, and participation in a community of sexual minorities [54]. In this regard, some organizations in Peru such as “Más igualdad Perú”, “Promosex”, “Movimiento Homosexual de Lima”, “International Association of Gays and Lesbians”, and government institutions such as the “Defensoría del Pueblo” are entities that work in favor of this community and could be key pieces to promote the expression without fear of sexual orientation and gender identity.’

### 4.2. Limitations and Strengths

The present study has some limitations. The survey on which it was based used a non-probabilistic convenience sampling, which implies that it is not necessarily representative of the Peruvian LGBTI population. Furthermore, as it is a cross-sectional study, a causal relationship cannot be established, and is a potential cause of the differences in the findings between our study and previous ones, because different designs were used. Both mental health problems and the expression of sexual orientation and/or gender identity were self-reported, therefore, there may be an information bias. In the case of self-reporting of mental health problems, there may be a memory bias. In addition, it is not possible to know if the mental health problems were previously diagnosed by a doctor or it they were a perception of the participant. The survey does not have a validated or standardized scale to assess the expression of sexual orientation and/or gender identity; and it did not have an explanation for each variable, so they could be over- or under-reported. Within mental health problems, anxiety and/or depression were asked about without making a distinction between the two disorders. Likewise, no distinction was made when the expression of sexual orientation and/or gender identity was asked. This could cause the association between the variables to change if they are studied independently. The exposure variable was evaluated with the question (Do you express your sexual orientation and/or gender identity without fear?). So, those who disapproved could not express themselves in any way or could only express themselves with fear, which can represent an information bias. Another important limitation is the impossibility of knowing if the perceived discrimination was related to the person’s LGBTI status or to other possible causes of discrimination, such as ethnicity or sociodemographic status.

Likewise, there is no certainty of knowledge of the subgroups within the sexual orientation and gender identity by the population, so there could be sub- or over-registration in each category. This limitation is relevant since it is possible that both mental health problems and the impact of their identity expression vary between subgroups. Furthermore, although it may be tempting to analyze our results by subgroups, within a cross-sectional study such as ours, the number of individuals analyzed would be significantly reduced, thus losing the power of the association [55]. Similarly, the analysis of a high number of subgroups (in our study we have five and six subgroups within sexual orientation and gender identity, respectively) would increase the risk of finding differences that do not really exist [56]. In addition, we could not know if the possible differences found between subgroups were real or if it would be a baseline effect of the individuals [56].

The survey was only available in Spanish, online, and the participants were reached primarily through LGBTI organizations. This could mean that people who did not belong to LGBTI organizations, did not have access to the Internet, spoke another language, or belonged to ethnic minorities did not participate. In addition, although the origin implies different characteristics that can influence our association, the origin was not significant in the bivariate analysis, probably because most of the respondents were residents of Lima. Finally, there are factors that could intervene in the development of mental health problems in the LGBTI population that were not measured in the survey, such as internalized heterosexism, self-acceptance [19], victimization, substance abuse, stigma, bullying, intrapersonal skills, and emotional regulation [57].

Among the strengths, we can point out that the main LGBTI organizations in Peru accompanied the process of designing the questionnaire and the pilot test [3] and it is the database that currently has the largest sample nationwide. Despite these limitations, this study is the first to link the expression of sexual orientation and/or gender identity with mental health problems in the LGBTI population in a middle-income country such as Peru.

## 5. Conclusions and Recommendations

### 5.1. Conclusions

In conclusion, more than half of the members of the LGBTI community do not express their sexual orientation and/or gender identity, and close to one in four have perceived mental health problems. The LGBTI population who expressed their sexual orientation and/or gender identity were less likely to have mental health problems.

We consider that the main contribution of our research is to highlight the possible implication that the expression of sexual orientation and/or gender identity has on the mental health of the LGBTI community in Peru. In this way, we suggest implementing public policies at the health level that promote the expression of sexual orientation and gender identity in favor of the mental health of the LGBTI community.

### 5.2. Recommendations

We suggest that future surveys evaluate the mental health and expression of the LGBTI population including validated and standardized scales, in order to distinguish mental health problems, the expression of sexual orientation, and the expression of gender identity separately. The latter because mental health problems and forms of expression can differ in the same individual.

Likewise, we suggest that the surveys should be more inclusive, with the probability of including languages other than Spanish, since they represent the diversity of the population in our country [58]. Furthermore, we suggest approaching the population through other forms of access besides the Internet, due to the known access problems in our territory [59]. Finally, it would be convenient to try to reach the target population not only through LGBTI organizations to improve the representativeness of the selected sample.

## Figures and Tables

**Figure 1 ijerph-20-05655-f001:**
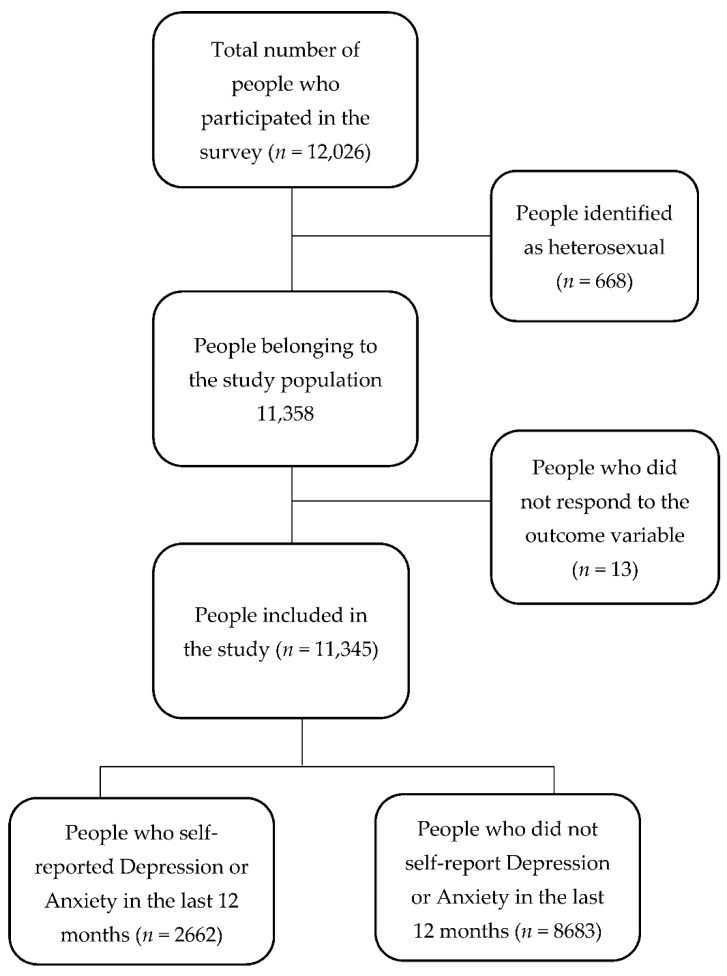
Participants flow chart.

**Table 1 ijerph-20-05655-t001:** Sociodemographic characteristics of the LGBTI population in Peru.

Characteristics	Frequency	Percentage (%)	95% CI
Sex assigned at birth			
Female	5156	46.89	(46.96,47.83)
Male	5839	53.11	(52.15,54.04)
Sexual orientation			
Gay	4749	43.2	(42.27,44.13)
Lesbian	2468	22.45	(21.67,23.24)
Bisexual	2862	26.03	(25.21,26.87)
Pansexual	578	5.26	(4.85,5.69)
Asexual	75	0.68	(0.56,0.85)
Other	261	2.37	(2.09,2.68)
Gender Identity			
Female trans	224	2.06	(1.80,2.34)
Male trans	231	2.12	(1.86,2.41)
Gender queer	796	7.32	(6.84,7.83)
No trans	9606	88.36	(87.75,88.96)
Other	14	0.13	(0.07,0.21)
Fearless expression of sexual orientation and/or gender identity			
Yes	4550	41.41	(40.49,42,34)
No	6437	58.59	(57.66,59.51)
Relationship status			
Without partner	5331	48.78	(47.83,49.72)
With partner	5443	49.8	(48.86,50.74)
With more than one partner	155	1.42	(1.20,1.65)
Respect, acceptance, and family integration			
Yes	3934	46.16	(45.09,47.22)
No	4589	53.84	(52.73,54.91)
Discrimination			
Yes	7160	70	(69.10,70.88)
No	3069	30	(29.11,30.90)
Perceived mental health problems			
Yes	2662	23.46	(22.69,24.26)
No	8683	76.54	(75.74,77.31)

**Table 2 ijerph-20-05655-t002:** Bivariate analysis according to the reported mental health problems.

Characteristics	Report Perceived Mental Health Problems						
	No			Yes			
	Frequency	Percentage (%)	95% CI	Frequency	Percentage (%)	95% CI	*p*-Value
Age	8491	25 (22–30) *	(25,26)	2588	23 (20–28) *	(23,23)	*p* < 0.001 **
Place of birth							
Lima	4682	59.64	(57.88,61.82)	1457	59.86	(58.55,60.73)	*p* = 0.849
Other	3168	40.36	(38.18,42.11)	977	40.14	(39.27,41.45)	
Place of origin							
Lima	6043	69.64	(66.79,70.36)	1826	68.6	(68.66,70.61)	*p* = 0.304
Other	2634	30.36	(29.64,33.20)	836	31.4	(29.39,31.34)	
Education							
Uneducated	18	0.21	(0.12,0.33)	8	0.3	(0.13,0.60)	*p* < 0.001
Initial	4	0.05	(0.01,0.11)	2	0.08	(0.01,0.27)	
Primary	19	0.22	(0.13,0.34)	6	0.23	(0.08,0.49)	
Secondary	1049	12.12	(11.44,12.83)	397	14.98	(13.64,16.39)	
Superior	6657	76.92	(76.02,77.80)	2067	77.97	(76.34,79.53)	
Postgraduate	907	10.48	(9.84,11.15)	171	6.45	(5.54,7.45)	
Job							
Yes	4915	61.6	(60.52,62.67)	1254	50.02	(48.04,52.00)	*p* < 0.001
No	3064	38.4	(37.33,39.48)	1253	49.98	(0.48,0.52)	
Ethnicity							
White	1445	18.23	(17.39,19.10)	395	15.87	(14.46,17.37)	*p* < 0.05
Mixed race	5505	69.45	(68.43,70.47)	1756	70.55	(68.72,72.34)	
Ethnical minorities	976	12.31	(11.60,13.06)	338	13.58	(12.26,14.99)	
Disability							
Yes	164	1.97	(1.68,2.29)	175	6.74	(5.81,7.78)	*p* < 0.001
No	8170	98.03	(97.71,98.32)	2420	93.26	(92.22,94.19)	
Health insurance							
Yes	6276	72.38	(71.42,73.32)	1738	65.29	(63.45,67.10)	*p* < 0.001
No	2395	27.62	(26.68,28.58)	924	34.71	(32.90,36.55)	
Sex assigned at birth							
Female	3748	44.65	(43.58,45.72)	1408	54.13	(52.20,56.06)	*p* < 0.001
Male	4646	55.35	(54.28,56.42)	1193	45.87	(43.94,47.80)	
Sexual orientation							
Gay	3840	45.76	(44.69,46.83)	909	34.85	(33.11,36.82)	*p* < 0.001
Lesbian	1987	23.68	(22.77,24.60)	481	18.49	(17.02,20.04)	
Bisexual	2006	23.9	(22.99,24.83)	856	32.91	(31.11,34.75)	
Pansexual	326	3.88	(3.48,4.32)	252	9.69	(8.58,10.89)	
Asexual	40	0.48	(0.34,0.65)	35	1.35	(0.94,1.87)	
Other	193	2.3	(1.99,2.64)	68	2.61	(2.04,3.30)	
Gender Identity							
Female trans	168	2.03	(1.72,2.35)	56	2.17	(1.64,2.80)	*p* < 0.001
Male trans	173	2.09	(1.79,2.42)	58	2.25	(1.71,2.90)	
Gender queer individuals	530	6.39	(5.88,6.96)	266	10.31	(9.16,11.54)	
Non-transsexual	7409	89.37	(88.69,90.03)	2197	85.12	(83.69,86.47)	
Other	10	0.12	(0.06,0.22)	4	0.15	(0.04,0.40)	
Fearless expression of sexual orientation and/or gender identity							
Yes	3640	43.41	(42.34,44.47)	910	34.99	(33.15,36.85)	*p* < 0.001
No	4746	56.59	(55.53,57.66)	1691	65.01	(63.15,66.85)	
Relationship status							
Without partner	3885	46.62	(45.54,47.69)	1446	55.7	(53.76,57.62)	*p* < 0.001
With partner	4339	52.07	(50.99,53.14)	1104	42.53	(40.61,44.45)	
With more than one partner	109	1.31	(1.07,1.57)	46	1.77	(1.30,2.35)	
Respect, acceptance, and family integration							
Yes	3177	48.33	(47.11,49.54)	757	38.84	(36.67,41.05)	*p* < 0.001
No	3397	51.67	(50.46,5289)	1192	61.16	(58.95,63.33)	
Discrimination							
Yes	5203	66.89	(65.83,67.93)	1957	79.88	(78.23,81.45)	*p* < 0.001
No	2576	33.11	(31.81,33.91)	493	20.12	(18.55,21.77)	

* Mediana (Rango intercuartílico); ** U de Mann–Whitney.

**Table 3 ijerph-20-05655-t003:** Association between the expression of sexual orientation and/or gender identity and mental health.

Characteristics	Sample Size = 11,345					
	Crude Model ^a^			Adjusted Model ^b^		
	cRP	95% CI	*p*-Value	aRP	95% CI	*p*-Value
Fearless expression of sexual orientation and/or gender identity						
No	Ref					
Si	0.76	(0.71,0.82)	*p* < 0.001	0.83	(0.76,0.90)	*p* < 0.001

cPR: crude prevalence ratio, aPR: adjusted prevalence ratio, 95% CI: 95% confidence interval. Ref: reference category; ^a^ generalized linear model Poisson family with robust variance; ^b^ generalized linear model Poisson family with robust variance adjusted for employment, disability, insurance, sexual orientation, respect, acceptance, and family integration, relationship status and discrimination.

## Data Availability

The data that support the findings of this study are openly available in the virtual platform, microdata section of the National Institute of Statistics and Informatics (INEI) at http://iinei.inei.gob.pe/microdatos/Consulta_por_Encuesta.asp (accessed on 1 December 2022). These data were obtained from the “query by survey” section, entering in Survey: First virtual survey for LGBTI people in Peru, in year: 2017 and in period: annual.

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
