# Peer review of "Association between the Expression of Sexual Orientation and/or Gender Identity and Mental Health Perceptions in the Peruvian LGBTI Population"

_ijerph, 2023, doi:10.3390/ijerph20095655_

Round 1
Reviewer 1 Report
I consider this to be a well conducted and well presented study.
I have 2 minor suggestions:
1. The English is of reasonable standard, but improvement would make the paper much more readable.
2. In the limitations section, it would be useful to comment on the limitations of the cross-sectional nature of the study in terms of drawing inference. Differences in study design may in part account for differences in findings across studies. This relates to, but expands upon, the point you make already about convenience sampling.
Reviewer 2 Report
Association between the expression of sexual orientation and/or gender identity and mental health in the Peruvian LGBTI population
This in an interesting, important, and well written contribution with relevant contributions to public health and inclusive interventions. The main problem with this article is that all measures are not systematically or psychometrically validated, and are based on dichotomic categorical variables. This doesn’t prevent its publication, but mental health must be properly described, since it was not assessed with valid measures. I would suggest authors to change the title to “mental health perceptions”.
Also, other changes would improve the overall quality of the article and improve its chances to be published, namely, the inclusion of more information in the abstract, specifically, what measures were used to assess mental health; and study objectives must be clearly and specifically stated.
Best wishes.
